# Efficient LLM Architectures

## Abstract

Recent LLMs have hundreds of billions of parameters consuming vast resources. Furthermore, the so called "AI scaling law" for transformers suggests that the number of parameters must scale linearly with the size of the data. In response, we inquire into efficient LLMs, i.e. those with the fewest parameters that achieve the desired accuracy on a training corpus. Specifically, by comparing theoretical and empirical estimates of the Kullback-Leibler divergence, we derive a natural AI scaling law that the number of parameters in an efficient LLM scales as $D^\gamma$ where $D$ is the size of the training data and $\gamma \in [0.44, 0.72]$, suggesting the existence of more efficient architectures. Against this backdrop, we propose recurrent transformers, combining the efficacy of transformers with the efficiency of recurrent networks, progressively applying a single transformer layer to a fixed-width sliding window across the input sequence. Recurrent transformers (a) run in linear time in the sequence length, (b) are memory-efficient and amenable to parallel processing in large batches, (c) learn to forget history for language tasks, or accumulate history for long range tasks like copy and selective copy, and (d) are amenable to curriculum training to overcome vanishing gradients. In our experiments, we find that recurrent transformers perform favorably on benchmark tests.

## 1 Introduction

LLMs are artificial neural networks typically built on the transformer architecture as proposed by Vaswani et al. (2017). Recent LLMs have hundreds of billions of parameters consuming vast amount of resources, e.g.,Brown et al. (2020), Rae et al. (2021), Smith et al. (2022), Thoppilan et al. (2022). Furthermore, the so called "AI scaling law" for transformers suggests that the number of parameters must scale linearly with the size of the data, and that we are running out of usable training data to build even larger models, e.g., Kaplan et al. (2020), Hoffmann et al. (2022), Muennighoff et al. (2023). In response, we inquire into efficient LLMs independent of architecture, i.e. those with the fewest parameters that achieve the desired accuracy on a training corpus.

We compare theoretical and empirical estimates of the Kullback-Leibler divergence, Kullback & Leibler (1951), to show that at prevalent sequence lengths, the number of unique sequences in a natural training corpus of size $D$ is between $\Omega(D^{0.44})$ and $\mathcal{O}(D^{0.72})$. Our result leverages the empirical estimates of Hoffmann et al. (2022). From a learning theoretic perspective, the number of unique sequences as a function of the size of a training corpus is an upper bound on the dimension of the space of functions required to learn any natural data set of that size. Theoretically, the number of parameters in an efficient LLM would scale linearly with the dimension of the space, and hence linearly with the number of unique sequences. In contrast to the so-called AI scaling law for the transformer architecture that suggests the number of parameters must scale linearly with the size of the data, a salient implication of our result is that the number of parameters in an efficient LLM scales as $D^\gamma$ for $0.44 \le \gamma \le 0.72$, suggesting the existence of more efficient LLM architectures.

Our result also has implications for so-called emergent abilities i.e. *"abilities that are not present in smaller-scale models but are present in large-scale models; thus they cannot be predicted by simply extrapolating the performance improvements on smaller-scale models"* e.g., Wei et al. (2022), Ganguli et al. (2022), Arora & Goyal (2023) and Srivastava et al. (2023). Conversely, Schaeffer et al. (2023) find that LLMs may exhibit abilities that are latent but not new when the number of parameters is increased. Bridging the above, our result suggests that if the number of parameters of an LLM is smaller than the number of unique sequences in the training corpus, scaling up can uncover latent abilities.

With our theoretical results as backdrop, we seek more efficient LLM architectures. Preceding transformers, Recurrent Neural Networks (RNNs), e.g., Elman (1990), their variants Long Short-Term Memory (LSTM), e.g., Hochreiter & Schmidhuber (1997) and Gated Recurrent Units (GRUs), e.g., Cho et al. (2014), were popular for linear time processing of sequential data streams such as natural language processing. More recent architectural alternatives include State Space Models, e.g., Gu et al. (2021), and minimal RNNs, Feng et al. (2024). Building on prior work, we propose recurrent transformers, combining the efficacy of transformers with the efficiency of recurrent networks, progressively applying a single transformer layer to a fixed-width sliding window across the input sequence. Prior work on recurrent transformers, e.g., Dai et al. (2019), where recurrence is used to extend context but does not participate in gradient calculations; and Hutchins et al. (2022), focuses on improving the long-range performance of multi-layer transformers by making some of the layers recurrent with large block sizes and many additional weights. In contrast, our focus is to find small and efficient networks for a given task, in the form of a single recurrent layer with a scalar parameter that learns to accumulate or forget history. Specifically, our recurrent transformers (a) run in linear time in the sequence length; (b) are memory-efficient and amenable to parallel processing in large batches (c) learn to forget history for language tasks, or accumulate history for long-range tasks like copy and selective copy; and (d) are amenable to curriculum training to overcome vanishing gradients, e.g.,Hochreiter (1991), Sodhani et al. (2020). In our experiments, we find that a single-layer recurrent transformer can match the performance of multi-layer transformers on benchmark tests, delivering comparable accuracy with fewer parameters and much lower computational cost. Furthermore, the trainable accumulation capability enables a recurrent transformer to solve long range tasks at a fraction of the size of other recurrent architectures, e.g., Gu & Dao (2023), Ren et al. (2024), and at a fraction of the cost of regular transformers.

## 2    BACKGROUND

An LLM is a statistical machine that takes as input a sequence of words of specified maximum length on a specified alphabet, and produces as output a probability distribution on possible next words. Training involves adjusting the model's parameters while running through a body of text called the training corpus, to minimize the statistical error between the model and the corpus. The error in the model, also known as the loss function, is chosen as a continuous function to enable the use of stochastic gradient descent for adjusting the parameters of the model. The commonly used loss function involves the entropy of an infinite stream of words, as defined by Shannon (1948), Cover (1999). Intuitively, entropy is the measure of information in the stream and is a lower bound on the capacity of a channel required to transmit the stream.

Let $l > 0$ specify the sequence length. Consider a sequence of $l$ words, $s = w_1 w_2 ... w_{l-1} w_l$. Let $\hat{s} = w_1 w_2 ... w_{l-1}$ denote the prefix of the first $l-1$ words of $s$. Given $\hat{s}$, humans can predict the last word $w_l$ of $s$ according to a conditional probability distribution $P(s) = P(w_l | \hat{s})$. Likewise, an LLM consumes the prefix $\hat{s}$ and computes a conditional probability distribution $Q(s) = Q(w_l | \hat{s})$ for the last word $w_l$. Let $\mathbb{P}\{x\}$ be the probability that a sequence $x$ occurs in the training corpus $T$, and let

$$p_s = \mathbb{P}\{\hat{s}\} P(s) = \mathbb{P}\{w_1 w_2 ... w_{l-1}\} P(w_l | w_1 w_2 ... w_{l-1})$$
$$q_s = \mathbb{P}\{\hat{s}\} Q(s) = \mathbb{P}\{w_1 w_2 ... w_{l-1}\} Q(w_l | w_1 w_2 ... w_{l-1}) \tag{1}$$

Let $S$ be the set of all sequences of length $l$ in the training corpus $T$. The Kullback-Leibler divergence of an LLM is

$$\Delta = \sum_{s \in S} p_s log(p_s/q_s) = \sum_{s \in S} p_s log(p_s) - \sum_{s \in S} p_s log(q_s) \tag{2}$$

The Kullback-Leibler divergence is well established to be non-negative. Rearranging, we get

$$-\sum_{s \in S} p_s log(q_s) = -\sum_{s \in S} p_s log(p_s) + \Delta$$

The term on the left above is the cross-entropy loss of the LLM on the training corpus. The first term on the right converges to the inherent entropy of the natural language on the training corpus. During the training process, the parameters of the LLM are iteratively adjusted to minimize the cross-entropy. In doing so, the LLM is effectively minimizing the Kullback-Leibler divergence, since the inherent entropy of the language is constant.

**Assumption 1.** *All calculations are performed in finite precision such that probabilities $p_s$ and $q_s$ are quantized over a discrete set of finitely many values $Y \subset [0, 1]$, and to avoid overflow in loss calculations the logarithm function is capped, i.e., for all $p, q \in Y$, $|log(p/q)| \leq \lambda$, for some positive constant $\lambda$.*

## 3 EMPIRICAL SCALING

Recent prior works report empirical expressions for the relationship between training loss and model scale, e.g., Kaplan et al. (2020), Hoffmann et al. (2022), Muennighoff et al. (2023). Per current practice, e.g., Hoffmann et al. (2022), Xue et al. (2023), these works train LLMs on a single epoch of natural training corpora. Specifically, Hoffmann et al. (2022) finds that for fixed sequence length, across a range of LLMs with $N$ parameters and training corpus of size $D$, the Kullback-Leibler divergence scales as follows

$$\Delta = \frac{A}{N^\alpha} + \frac{B}{D^\beta}, \tag{3}$$

where $\alpha = 0.34$, $\beta = 0.28$, $A = 406.4$ and $B = 410.7$ are fitting constants.

In the experiments, the training corpus is randomly split into a training subset and a test subset. The training subset is used for training the LLM, and the Kullback-Liebler divergence is computed on the test subset. Since each training run starts with a random initialization of the neural network, we assume Equation (3) holds with high confidence $(1 - \delta)$, for small positive $\delta$ . The first term on the right reflects the ability of an LLM with $N$ parameters to capture the richness of the training corpus. The second term reflects the ability of the training corpus to capture the richness of the natural language. Since $\alpha \approx \beta, A \approx B$, and the compute cost is $N \times D$, it follows that the optimal scaling strategy is $N \propto D$, which is also known as the AI Scaling Law. However, $N \propto D$ suggests rote learning, raising the question of whether it is a natural requirement or a reflection of the transformer architecture used in the experiments.

## 4 THEORETICAL SCALING

We now consider LLMs in the context of the theoretical model of Probably Approximately Correct (PAC) learning, Valiant (1984). Recall that $S$ is the set of sequences of length $l$ in $T$. Consider a hypothesis space of functions $F = \{f : S \to Y\}$, where for $s \in S$, $f(s)$ is a hypothesis $q_s$ for the probabilities $p_s$ of Equation 1. We follow Shalev-Shwartz & Ben-David (2014) in defining an agnostic learning algorithm as below. As in Section 3, the training corpus $T$ provides $D$ sequences of length $l$ as training samples for the learning algorithm. We define the loss function of the learning algorithm to be the Kullback-Leibler divergence between $f \in F$ and the natural distribution $p_s$ as follows

$$L(f, T) = \sum_{s \in S} p_s log(p_s/f(s))$$

**Definition 1.** *Given $\epsilon, \delta \in (0, 1)$ an agnostic learning algorithm finds $f \in F$ with confidence at least $(1 - \delta)$ such that*

$$L(f, T) \leq \min_{h \in F} L(h, T) + \epsilon$$

We assume that the natural distribution follows Zipf's law, Zipf (2013) for natural languages, which is widely established and valid for sequences, Ryland Williams et al. (2015).

**Assumption 2.** *The natural distribution follows Zips's law in that $p_s \propto \frac{1}{z+Z}$, where $z$ is the rank of $s$ in descending order of frequency amongst $s \in S$, and $Z$ is a constant.*

The following theorem uses the asymptotic complexity notations $\mathcal{O}$ and $\Omega$ for the upper and lower bound respectively.

**Theorem 1.** *For any agnostic learning algorithm for $F$, $\epsilon$ is $\Omega(|S|/D)$ and $\mathcal{O}[(|S|/D)^{0.5}]$*

*Proof.* We first prove the upper bound. Let $\hat{p}_s$ and $\hat{L}(f, T)$ be the observed probabilities and loss respectively in a randomly chosen training subset of the corpus $T$. We have

$$|\hat{L}(f, T)| = |\sum_{s \in S} \hat{p}_s log(\hat{p}_s/f(s))| \leq \sum_{s \in S} |\hat{p}_s log(\hat{p}_s/f(s))| = \sum_{s \in S} \hat{p}_s |log(\hat{p}_s/f(s))|$$

Invoking Assumption 1 in the above, we get

$$|\hat{L}(f,T)| \leq \sum_{s \in S} \hat{p}_s |log(\hat{p}_s/f(s))| \leq \sum_{s \in S} \hat{p}_s \lambda \leq \lambda$$

By Hoeffding's inequality, Feller (1991), we get

$$\mathbb{P}\left\{\left|\hat{L}(f,T) - L(f,T)\right| > \epsilon\right\} \leq 2e^{-2D\epsilon^2/\lambda}$$

Summing over all choices for $f$, we get

$$\sum_{f \in F} \mathbb{P}\left\{\left|\hat{L}(f,T) - L(f,t)\right| > \epsilon\right\} \leq |Y|^{|S|} 2e^{-2D\epsilon^2/\lambda}$$

We wish to bound $\delta$ with the RHS above, i.e., $\delta \leq |Y|^{|S|} 2e^{-2D\epsilon^2/\lambda}$. Taking logarithms on both sides and treating $\delta, \lambda$, and $|Y|$ as constants yields $\epsilon = \mathcal{O}[(|S|/D])^{0.5}$.

Next we prove the lower bound. For any $f \in F$,

$$L(f,T) - \hat{L}(f,T) = \sum_{s \in S} p_s log(p_s/f(s)) - \sum_{s \in S} \hat{p}_s log(\hat{p}_s/f(s))$$

$$= \sum_{s \in S} p_s log(p_s/\hat{p}_s) - \sum_{s \in S}(\hat{p}_s - p_s)log(\hat{p}_s) + \sum_{s \in S}(\hat{p}_s - p_s)log(f(s))$$

Taking the expectation $\mathbb{E}$ on both sides of the above with respect to random training subsets of T, and noting that $\mathbb{E}\{(\hat{p}_s - p_s)\} = 0$, we get,

$$\mathbb{E}\left\{L(f,T) - \hat{L}(f,T)\right\} = \mathbb{E}\left\{\sum_{s \in S} p_s log(p_s/\hat{p}_s)\right\} \tag{4}$$

Since each sequence $s$ is a Bernoulli variable in a random training subset, the variance of $\hat{p}_s$ is $p_s(1 - p_s)$. Therefore by the Central Limit Theorem, e.g., Feller (1991), as $D$ increases without bound, almost surely,

$$|\hat{p}_s - p_s| \propto \sqrt{\frac{p_s(1 - p_s)}{D}}$$

Invoking Assumption 2 in the above and summing over $s \in S$, we get

$$\sum_{s \in S} |\hat{p}_s - p_s| \propto \frac{1}{\sqrt{D}} \sum_{z=1}^{|S|} \sqrt{\frac{1}{z + Z}\left(1 - \frac{1}{z + Z}\right)}$$

Approximating the sum in the right-hand-side above with the definite integral for $S \gg Z$, we get that

$$\sum_{s \in S} |(\hat{p}_s - p_s| \propto \sqrt{\frac{|S|}{D}} \tag{5}$$

Combining the above with Pinsker's inequality, Csiszár & Körner (2011), we get

$$\sum_{s \in S} p_s log(p_s/\hat{p}_s) \geq \frac{1}{2}\left(\sum_{s \in S} |(\hat{p}_s - p_s|\right)^2 = \Omega[|S|/D]$$

Combining the above with Equation 4, we set

$$\zeta = \mathbb{E}\left\{L(f,T) - \hat{L}(f,T)\right\} = \Omega[|S|/D]$$

By Assumption 1, $L(f,T) - \hat{L}(f,T) \leq 2\lambda$. Let $x$ be the probability that $L(f,T) - \hat{L}(f,T) > \zeta/2$. The minimum value of $x$ is achieved by placing a probability mass of $x$ at $2\lambda$ and $(1 - x)$ at $\zeta/2$. Hence

$$2\lambda x + (\zeta/2)(1 - x) = \zeta \implies x(2\lambda - \zeta/2) = \zeta/2 \implies x = \frac{\zeta}{4\lambda - \zeta} > \frac{\zeta}{4\lambda}$$

It follows that with probability greater than $\frac{\zeta}{4\lambda}$,

$$L(f, T) - \hat{L}(f, T) > \zeta/2 = \Omega[|S|/D]$$

implying that for $f \in F$ output by a learning algorithm, $L(f, T) > \min_{h \in F} L(h, T) + \Omega[|S|/D]$. Hence the theorem.

$\square$

**Remark**: It is easy to see that Equation 5, and therefore Thereom 1 also hold when the natural distribution is the uniform distribution, i.e. $p_s = 1/|S|$ and $|S| \gg 1$.

We now link the empirical bounds of Equation 3 with the theorem above to estimate the number of unique sequences in a training corpus as a function of its size. Our analysis compares the empirical estimates of Hoffmann et al. (2022) with the estimates of Theorem 1 to extract properties of the underlying natural language.

**Lemma 1.** *If Equation 3 holds for sequences of length $l$, the number of unique sequences $|S|$ of length $l$ in a corpus of size $D$ is between $\Omega[D^{0.44}|]$ and $\mathcal{O}[D^{0.72}]$*

*Proof.* Consider an LLM with $N$ parameters representing a space of functions $F = \{f : S \to Y\}$. Suppose the LLM computes a function $f$ after training on corpus $T$. From Equation 2 and the upper bound of Theorem 1, we have

$$\Delta = \sum_{s \in S} p_s log(p_s/q_s) = \min_{h \in F} L(h, T) + \mathcal{O}[(|S|/D)^{0.5}]$$

Combining Equations 3 with the above, we get

$$\Delta = \frac{A}{N^\alpha} + \frac{B}{D^\beta} = \min_{h \in F} L(h, T) + \mathcal{O}[(|S|/D)^{0.5}]$$

As the number of parameters $N \to \infty$, $A/N \to 0$, and since transformers are universal approximators, $\min_{h \in F} L(h, T) \to 0$. Hence we have

$$\frac{B}{D^\beta} = \mathcal{O}[(|S|/D)^{0.5}]$$

Substituting $\beta = 0.28$ from Equation 3 in the above, we get

$$|S| = \Omega[D^{1-2\beta}] = \Omega[D^{0.44}].$$

Next, we combine Equations 2 with the lower bound of Theorem 1.

$$\Delta = \sum_{s \in S} p_s log(p_s/q_s) = \min_{h \in F} L(h, T) + \Omega[|S|/D]$$

Combining Equations 3 with the above, we get

$$\Delta = \frac{A}{N^\alpha} + \frac{B}{D^\beta} = \min_{h \in F} L(h, T) + \Omega[|S|/D]$$

As the number of parameters $N \to \infty$, $A/N \to 0$, and since transformers are universal approximators, $\min_{h \in F} L(h, T) \to 0$. Hence we have

$$\frac{B}{D^\beta} = \Omega[|S|/D]$$

Substituting $\beta = 0.28$ from Equation 3 in the above, we get

$$|S| = \mathcal{O}[D^{1-\beta}] = \mathcal{O}[D^{0.72}].$$

Which completes the proof.

$\square$

From a learning theoretic perspective, the number of unique sequences as a function of the size of a training corpus is an upper bound on the degrees of freedom required to learn any natural dataset of that size. Theoretically, the number of parameters in an efficient LLM would scale linearly in the number of unique sequences; e.g., a "brute-force" code book of one entry per unique sequence. The so-called AI scaling law for the transformer architecture suggests that the number of parameters must scale linearly with the size of the data, e.g., Kaplan et al. (2020), Hoffmann et al. (2022), Muennighoff et al. (2023). In contrast, our main result implies that the number of parameters of an efficient LLM need only scale as $D^\gamma$ for $\gamma \in [0.44, 0.72]$, suggesting the existence of substantially more efficient LLM architectures.

Our result also has implications for so-called emergent abilities i.e. *"abilities that are not present in smaller-scale models but are present in large-scale models; thus they cannot be predicted by simply extrapolating the performance improvements on smaller-scale models"* e.g., Wei et al. (2022), Ganguli et al. (2022), Arora & Goyal (2023) and Srivastava et al. (2023). Conversely, Schaeffer et al. (2023) find that LLMs may not exhibit new abilities simply by increasing the number of parameters. Our result implies that if the number of parameters of an LLM is smaller than the number of unique sequences in the training corpus, scaling up the LLM can uncover emergent skills.

## 5 RECURRENT TRANSFORMERS

With our theoretical results as backdrop, we propose an architecture that converts any transformer into a recurrent network in order to combine the efficacy of the attention mechanism in the transformer, with the efficiency of recurrent networks. Specifically, let $X = (x_1, ...x_i, ...x_t)$ be an input sequence, where each $x_i$ is a block of vectors embedding the corresponding sequence of non-overlapping and contiguous input tokens $w_{i,1}, w_{i,2}, ...w_{i,k}$, with $k \geq 1$ being the block size. Let $\tau$ denote the function computed by a regular transformer, e.g., Vaswani et al. (2017), in that on input sequence $X$ the transformer outputs $\tau(X) = (y_1, y_2, ...y_t)$. We assume that the $x_i$ and $y_i$ consist of vectors of the same dimension, and that $\tau$ preserves causality in that $y_i$ does not depend on $x_{i+1}, x_{i+2}...x_t$. The recurrent transformer based on $\tau$ is as follows, where the comma between two sequences concatenates them:

$$h_1 = \tau(x_1) \tag{6}$$
$$(y_i, h_{i+1}) = \tau(\alpha h_{i-1} + h_i, \alpha h_{i-1} + x_{i+1}), \ \ \alpha \in [0, 1] \tag{7}$$
$$y_t = \tau(h_t) \tag{8}$$

In the above, $h_i$ and $y_i$ are respectively the hidden state and output of the recurrent transformer at position $i$. Equation 6 initializes the recurrence at position 1, Equation 7 iteratively applies the transformer function with causal attention between position $i+1$ and position $i$, and Equation 8 closes the recurrence at the last position. The trainable accumulation parameter $\alpha$ serves to accumulate a weighted portion of the hidden state $h_{i-1}$ into the working states of the network. Intuitively, $\alpha = 1$ favors memorization, while $\alpha = 0$ favors generalization. As we will see in our experiments, $\alpha \approx 0.5$ aids in long-range tasks. While the above architecture can support multiple layers stacked successively, we found no benefit to doing so in our experiments. Furthermore, we found that a small block size of $k \approx 32$ solved even long range tasks with ease.

### 5.1 COMPUTATIONAL COMPLEXITY

Let $r, d$ and $l$ be the number of layers, the embedding dimension, and the sequence length respectively of a regular transformer and $k$ the block size of a recurrent transformer. We note that the computational complexity of the regular transformer is $r(l^2d + ld^2)$, from which it is easy to see that of the recurrent transformer is $r(4dlk + 2ld^2)$.

## 6 EXPERIMENTAL RESULTS

We now test the performance of recurrent transformers on several datasets. The goal of our experiments is to test whether smaller single-layer recurrent transformers can match the performance of larger multi-layer transformers at lower computational cost. Owing to budge constraints, all of our

| $l$ | $d$ | Heads | Dropout |
|---|---|---|---|
| 1024 | 32 | 4 | 0.05 |

Table 1: Base Transformer Layer for Long Range Image Classification

| Model | Layers | Batch size | Parameters | Ops/sample | Val Loss | Val Acc. % |
|---|---|---|---|---|---|---|
| Regular Transformer | 4 | 40 | 9.1E4 | 1.38E8 | 1.68 (0.02) | 41.2 (0.7) |
| Regular Transformer | 1 | 40 | 5.3E4 | 3.46E7 | 1.75 (0.02) | 39.3 (0.8) |
| Recurrent Transformer | 1 | 120 | 5.3E4 | 6.29E6 | 1.50 (0.01) | 47.3 (0.4) |

Table 2: Experimental Results for Long Range Image Classification

experiments were run on an M4 Mac Mini with 16GB memory. All of our experiments use positional encoding of tokens and the AdamW optimizer.

## 6.1 Long Range Image Classification

We compare the performance of the recurrent transformer against regular transformers on the "Long Range Arena" image classification dataset of Tay et al. (2020):

*"This task is an image classification task, where the inputs are sequences of pixels. In other words, an $N \times N$ image is flattened to a sequence of length $N^2$ pixels. ...To simplify the setup, we map the input images to a single gray-scale channel where each pixel is represented with an 8-bit pixel intensity (vocabulary size of 256). In LRA, we use the CIFAR-10 dataset..."*

Specifically, the CIFAR-10 dataset, Krizhevsky et al. (2009), comprises a training set of 50,000 images of $32 \times 32$ pixels, and 10,000 test images across 10 categories. For this task, the images are converted into 256 grayscale values and fed as an input sequence of 1024 tokens across a vocabulary of 256. The output is a single token across a vocabulary of 10 items representing the classification label of the images. Table 1 specifies the base transformer layer in our comparison.

Table 2 shows the results of our experiments for this dataset, comparing a 4-layer and 1-layer regular transformer against a recurrent transformer with block size of 32. The "Ops/sample" column reflects our computational complexity estimates of section 5.1. The Validation Loss reported is the average across 100 samples and in parentheses, the corresponding standard error, e.g., 1.5 (0.01) implies that with 95% confidence, the validation loss is within $\pm$ 0.02 of 1.5. All three models were run for 600,000 samples, and the recurrent transformer could handle larger batch sizes than the 4-layer transformer in the available memory. The recurrent transformer outperforms on loss and accuracy, requiring less than 5% of the compute of the 4-layer regular transformer. Of interest, the parameter $\alpha$ of Equation 7 converges rapidly to $\approx 0.45$ during training, suggesting accumulation of history aids in solving this task. Figure 1 shows the loss and accuracy during training.

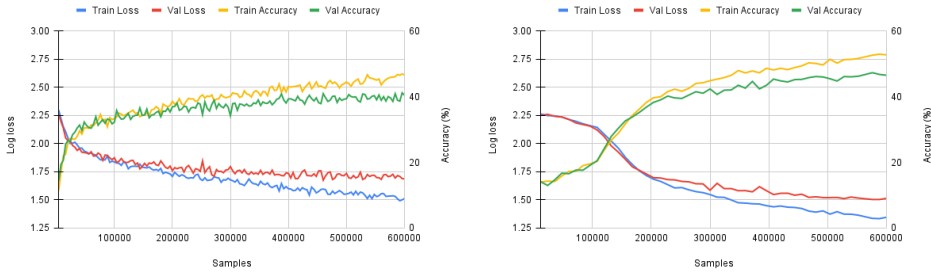

Figure 1: Long Range Image Classification: 4-layer regular (left) & recurrent transformer (right)

## 6.2 Copy and Selective Copy with Curriculum Training

We test the performance of the recurrent transformer on the copy and selective copy tasks of Gu & Dao (2023). The copy task involves 16 tokens, followed by 4096 noise tokens, followed by 16 tokens

| $l$ | $d$ | Heads | Dropout |
|---|---|---|---|
| 4128 | 96 | 6 | 0.05 |

Table 3: Base Transformer Layer for Copy & Selective Copy with $\approx 0.5M$ parameters

| Task | Batch Size | Ops/sample | $\alpha$ | Val Loss | Val Acc. % |
|---|---|---|---|---|---|
| Copy | 64 | 2.5E7 | 0.60 | 0.000 (3E-7) | 100 (0.00) |
| Selective Copy | 64 | 2.5E7 | 0.50 | 0.036 (3E-3) | 99.4 (0.04) |

Table 4: Experimental results for the recurrent transformer on Copy & Selective Copy

to trigger recall of the first 16 tokens, for a total sequence of 4128 tokens. The tokens are drawn from an alphabet of 16 unique characters, of which 2 characters are reserved for the noise and recall tokens respectively. Successful copy would reproduce the first 16 tokens in the original order against the last 16 recall tokens. The selective copy task randomly intersperses the 16 tokens to be copied amongst 4096 noise tokens, followed by 16 tokens to trigger recall of the 16 interspersed tokens, for a total sequence of 4128 tokens.. Successful copy would reproduce the 16 interspersed tokens in their original order against the last 16 recall tokens.

It is known that copy tasks are easy for transformers, but challenging for recurrent networks, requiring $\approx 100M$ parameters, e.g., Gu & Dao (2023), Ren et al. (2024). In our experiments, the surprisingly small recurrent transformer of Table 3 with 0.5M parameters solves the tasks, with block size 32. Training the recurrent transformer on the copy tasks is challenging due to vanishing gradients, e.g., Hochreiter (1991). We overcome this challenge with curriculum training, e.g., Sodhani et al. (2020). Specifically, the training process begins with say, 128 noise tokens, trains until the validation loss drops down to say 0.3, then doubles the number of noise tokens. This process is repeated until the number of noise tokens reaches 4096, at which point the training process drives the validation loss down to the desired minimal level for completion. During training, the learning rate decays as a function of the sequence length and the validation loss.

Table 4 shows the results of our experiments. The Ops/sample reported in the table is when each sample has 4096 noise tokens, which is the case during inference. The Validation Loss and Validation Accuracy are averaged over 100 samples and the numbers in parentheses are the respective standard error. For comparison, the estimated Ops/sample is $\approx 1\%$ of a regular transformer with four layers of Table 3. The values of the accumulation parameter $\alpha$ in the table suggest a balance between forget and accumulate to solve this task. Figure 2 shows the loss and accuracy for the selective copy task during the training process. The spikes in the figures are the result of the shocks introduced by doubling the number of noise tokens during curriculum training. The validation loss and accuracy are better than the corresponding training loss and accuracy due to the dropout during training.

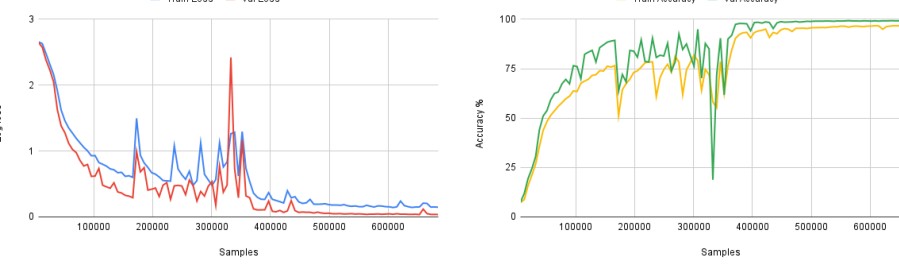

Figure 2: Curriculum training of recurrent transformer for Selective Copy

## 6.3 NATURAL LANGUAGE PROCESSING

We compare the performance of the recurrent transformer against regular transformers on the nanoGPT Shakespeare model and dataset of Karpathy (2023). Table 5 specifies the base transformer layer in our comparison. The recurrent transformer has a block size 16. Table 6 shows the results of our experiments comparing the performance of a 1-layer and 6-layer regular transformer operating on the

| $l$ | $d$ | Heads | Dropout |
|---|---|---|---|
| 256 | 384 | 6 | 0.2 |

Table 5: Base Transformer Layer for Shakespeare LLM

| Model | Layers | Batch size | Parameters | Ops/sample | Val Loss |
|---|---|---|---|---|---|
| Regular Transformer | 6 | 64 | 11E6 | 3.8E8 | 1.47 (0.01) |
| Regular Transformer | 1 | 64 | 1.E6 | 6.3E7 | 1.58 (0.01) |
| Recurrent Transformer | 1 | 64 | 1.9E6 | 8.2E7 | 1.47 (0.01) |

Table 6: Experimental Results for Shakespeare LLM

full sequence length, against a 1-layer recurrent transformer. The Validation Loss is averaged over 40 samples and the number in parentheses is the respective standard error. The Ops/sample entries reflect the computational complexity estimates of Section 5.1. Figure 3 shows the Training Loss and Validation Loss for our experiments against the total training cost per Section 5.1. Although a fifth of the size, the recurrent transformer matches the performance of the multi-layer regular transformer in this experiment, requiring more samples to train but at a lower cost per sample, and therefore about the same total training cost. However, the recurrent transformer requires only about $20\%$ the inference cost of the multi-layer regular transformer. Of interest, the parameter $\alpha$ of Equation 7 converges rapidly to zero during training. Owing to budge constraints, we are unable to test larger models.

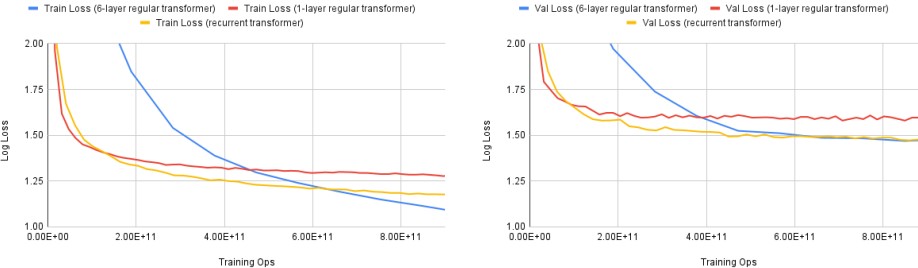

Figure 3: Shakespeare LLM: Training loss (left) and Validation Loss (right)

# 7 SUMMARY

Recent LLMs have hundreds of billions of parameters consuming vast resources. Furthermore, the so-called "AI scaling law" for transformers suggests that the number of parameters must scale linearly with the size of the data. In response, we inquire into efficient LLMs, i.e. those with the fewest parameters that achieve the desired accuracy on a training corpus. Specifically, by comparing theoretical and empirical estimates of the Kullback-Leibler divergence, we derive a natural AI scaling law that the number of parameters in an efficient LLM scales as $D^\gamma$ where $D$ is the size of the training data and $\gamma \in [0.44, 0.72]$, suggesting the existence of more efficient architectures. Against this backdrop, we propose recurrent transformers, combining the efficacy of transformers with the efficiency of recurrent networks, progressively applying a single transformer layer to a fixed-width sliding window across the input sequence. Recurrent transformers (a) run in linear time in the sequence length, (b) are memory-efficient and amenable to parallel processing in large batches, (c) learn to forget history for language tasks, or accumulate history for long range tasks like copy and selective copy, and (d) are amenable to curriculum training to overcome vanishing gradients. In our experiments, we find that recurrent transformers perform favorably on benchmark tests. In our experiments, we find that a single-layer recurrent transformer can match the performance of multi-layer transformers on benchmark tests, delivering comparable accuracy with fewer parameters and lower computational cost.

## 8 REPRODUCIBILITY

All code available as supplementary materials.

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
