# OpenReview forum: "Efficient LLM Architectures"
_ICLR.cc/2026/Conference — ICLR 2026 Conference Withdrawn Submission_

### Official Review · Reviewer_aV5S · 2025-10-27

**Soundness:** 2
**Presentation:** 3
**Contribution:** 2
**Rating:** 4
**Confidence:** 3

**Summary:**

This paper combines PAC learning theory and KL divergence equations to derive a new AI scaling law, which argues that the parameter count $N$ of an efficient LLM should be sublinear to the data size $D$. Based on this, the paper introduces an efficient architecture called the Recurrent Transformer. It adopts only one layer of Transformer, and the input is processed recurrently block by block. The historical hidden state is updated with a forget gate to enable forgetting, and the current block input is concatenated with the historical hidden state. Experiments show that the Recurrent Transformer excels over the standard Transformer with lower computational cost.

**Strengths:**

1. The paper derives a new AI scaling law arguing that the parameter count scales sublinearly with the data size, suggesting more efficient architectures. The theoretical analysis is useful and enlightening.
2. The new architecture, the Recurrent Transformer, is theory-driven and well-supported by the analysis above.
3. The Recurrent Transformer performs well on long-range copy tasks, which are a weakness for linear models.

**Weaknesses:**

1. There is a critical mismatch between the claims and the experimental scale. The paper derives a scaling law, but all the experiments involve very small-scale models (the maximum size is 11M). However, the emergent abilities of modern LLMs are considered to be something that only models with billions of parameters possess. A single-layer architecture with only 11M parameters outperforming a standard transformer provides little evidence that it is also effective at scales such as 1B, 7B, or larger.
2. The recurrent form of the transformer introduced in the paper is very similar to Transformer-XL[1]. Both combine the historical hidden state with the current input, and they both process the sequence block by block recurrently. Therefore, the Recurrent Transformer architecture may lack novelty.
3. The baselines are not sufficient. As the Recurrent Transformer has linear complexity, it would be better to compare it against other linear-complexity models such as Mamba and GLA.


[1] Transformer-XL: Attentive Language Models Beyond a Fixed-Length Context

**Questions:**

1. Can the authors provide any evidence or at least a reasonable argument to support that 'this single-layer architecture can maintain its effectiveness at the 1B+ parameter scale' ?
2. Why not choose Mamba or GLA as baselines? These models have extremely strong performance on long-sequence tasks and are also representatives of efficient architectures.

---

> ### Author Response · Authors · 2025-11-14
> **Thanks and response to review**
>
> Weaknesses:
>
> There is a critical mismatch between the claims and the experimental scale. The paper derives a scaling law, but all the experiments involve very small-scale models (the maximum size is 11M). However, the emergent abilities of modern LLMs are considered to be something that only models with billions of parameters possess. A single-layer architecture with only 11M parameters outperforming a standard transformer provides little evidence that it is also effective at scales such as 1B, 7B, or larger.
>
> We lack the resources for large scale experiments.   The paper does not claim that the architecture scales only that it shows promise.
>
> The recurrent form of the transformer introduced in the paper is very similar to Transformer-XL[1]. Both combine the historical hidden state with the current input, and they both process the sequence block by block recurrently. Therefore, the Recurrent Transformer architecture may lack novelty.
>
> This is a good point.  However, in contrast to our proposed architecture, the Transformer-XL does not calculate gradients on the hidden state that is carried forward.  Per Dai et al (2019):  "During training, the hidden state sequence computed for the previous segment is fixed and cached to be reused as an extended context when the model processes the next new segment, as shown in Fig. 2a. Although the gradient still remains within a segment, this additional input allows the network to exploit information in the history, leading to an ability of modeling longer-term dependency and avoiding context fragmentation."
> Our revised draft makes mention of this.
>
> The baselines are not sufficient. As the Recurrent Transformer has linear complexity, it would be better to compare it against other linear-complexity models such as Mamba and GLA.
>
> The proposed recurrent transformer outperforms the regular transformer which has quadratic complexity and is the prevalent LLM architecture in most practical applications.
>
> Thanks

---

> > ### Comment · Reviewer_aV5S · 2025-11-24
> >
> > Thanks for your reply. However, I think experiments on LLM with a scale larger than 11M is crucial for a paper claiming a "scaling law" which is just unavoidable due to limited computation resources. And other linear model baselines are important too. So I maintain my score.

---

> > > ### Author Response · Authors · 2025-11-28
> > > **Theorems are not claims**
> > >
> > > Regarding your comment:
> > > "However, I think experiments on LLM with a scale larger than 11M is crucial for a paper claiming a "scaling law" which is just unavoidable due to limited computation resources"
> > >
> > > This is a theoretical paper with illustrative experiments.  The Natural Scaling Law is not a "claim" but a theorem that is proven in the paper.
> > >
> > > The theorem stands for all models, not just linear ones.  And is independent of scale.

---

> ### Comment · Area_Chair_4GfW · 2025-11-23
>
> Dear reviewer,
>
> Thanks for your time and effort in reviewing ICLR2026 submissions. The authors have submitted their responses to your review. Please take the time to read and raise your further comments, and discuss with the authors.
>
> Best regards,
>
> AC

---

### Official Review · Reviewer_JEyk · 2025-10-28

**Soundness:** 2
**Presentation:** 2
**Contribution:** 3
**Rating:** 4
**Confidence:** 2

**Summary:**

The paper tackles an important problem and proposes an interesting architectural idea. The theoretical contribution is novel but needs strengthening. The experimental validation is insufficient for the claims made. With significant revisions addressing the theoretical gaps and experimental limitations, this could become a solid contribution.

**Strengths:**

1. Novel theoretical analysis: The connection between Kullback-Leibler divergence and unique sequence counts provides fresh perspective on parameter scaling

2. A Practical architecture: Recurrent transformers offer O(N) complexity vs O(N²) for standard transformers

3. Learnable accumulation parameter: The α parameter that learns to forget/accumulate history is elegant

**Weaknesses:**

1. Theoretical Issues

- Lemma 1 proof has logical gaps: The transition from dim(F) ≤ |S| to bounds on |S| relies on assumptions about optimal function spaces that aren't justified
- Assumption 1 is oversimplified: Finite precision with uniform quantization doesn't reflect actual neural network computation


2. Experimental Limitations

- **Very limited scale**: All experiments run on 16GB Mac Mini - cannot validate claims about large-scale efficiency
- **No comparison with recent efficient architectures**: Missing comparisons with Mamba, RWKV, RetNet, and other modern alternatives mentioned in related work
- **Cherry-picked baselines**: Comparing against "regular transformers" without positional encodings, modern optimizations

3. Presentation Issues

- Notation inconsistency: S used for both sequence set and individual sequences
- Missing details: How exactly is curriculum training scheduled? What are the learning rate schedules?
- Line 33: "sparse in that most parameters are negligible" - needs citation or evidence
- Line 270: Table reference formatting inconsistent
- Figure quality: Figures 1-3 have overlapping legends and are hard to read
- Related work: Missing discussion of recent efficient transformers, eg Mamba, RWKV, RetNet

**Questions:**

- Can you provide experiments at larger scale (>1B parameters) to validate the D^0.44-0.72 scaling law?
- Why does the theoretical analysis assume non-duplicative training corpora when real LLM training uses multi-epoch training?
- How does the recurrent transformer handle variable-length sequences during inference?
- What is the actual memory footprint comparison with baselines during training and inference?
- Can you provide ablation studies on:
    - Block size K
    - Number of layers
    - Impact of α initialization
- How does performance scale with sequence length beyond 4128 tokens tested?

---

> ### Author Response · Authors · 2025-11-14
> **Thanks and response to review**
>
> Theoretical Issues
>
>    (1)  Lemma 1 proof has logical gaps: The transition from $dim(F) \leq |S|$ to bounds on $|S|$ relies on assumptions about optimal function spaces that aren't justified
>
>    This is a good point. The revised draft steps through the proof in stages to close the gaps.
>
>    (2) Assumption 1 is oversimplified: Finite precision with uniform quantization doesn't reflect actual neural network computation
>
>    This is another good point.   The revised draft refines the assumption to allow for any quantization as long as log-loss overflow  is capped.
>
> Experimental issues
>
> Cherry picked baselines: Comparing against "regular transformers" without positional encodings, modern optimizations.
>
> All of our experiments use positional encodings and the AdamW optimizer in an "apples to apples" comparison.  The revised draft makes this is explicit.
>
> Questions:
> (1) Can you provide experiments at larger scale (>1B parameters) to validate the scaling law?
>
> Unfortunately this is outside of our capabilities.
>
> (2) Why does the theoretical analysis assume non-duplicative training corpora when real LLM training uses multi-epoch training?
>
> Excellent question.   As cited in the paper, the experiments of Hoffman et al (2022) are on single epochs of the training data since multiple epochs degrade LLM performance.  Seems common practice per the literature.
>
> (3) How does the recurrent transformer handle variable-length sequences during inference?
>
> All code is included in the supplementary material and supports variable length sequences during training or inference.
>
> (4) What is the actual memory footprint comparison with baselines during training and inference?
>
> We did not log such, but all code is posted for reproduction and analysis
>
> (5) Can you provide ablation studies on: Block size;  Number of layers; Impact of $\alpha$ initialization
>
> We lack the resources for larger studies
>
> (6) How does performance scale with sequence length beyond 4128 tokens tested?
>
> Unfortunately 4128 was the largest we could fit within our computational resources.
>
> Thanks

---

> > ### Comment · Reviewer_JEyk · 2025-11-24
> >
> > Thanks for the detailed response and revisions. The clarifications on multi-epoch training and variable-length sequences are helpful. However, I believe the lack of >1B scale validation and ablation studies significantly affects the work's credibility. I think this work has potential, but these gaps are substantial and my assessment will remain unchanged.

---

> ### Comment · Area_Chair_4GfW · 2025-11-23
>
> Dear reviewer,
>
> Thanks for your time and effort in reviewing ICLR2026 submissions. The authors have submitted their responses to your review. Please take the time to read and raise your further comments, and discuss with the authors.
>
> Best regards,
>
> AC

---

> ### Author Response · Authors · 2025-11-28
> **Credibility**
>
> "I believe the lack of >1B scale validation and ablation studies significantly affects the work's credibility"
>
> The credibility of theoretical results is independent of model scale and ablation studies.  The experiments are illustrative suggestions for future work by researchers with full scale compute budgets.

---

### Official Review · Reviewer_6W5n · 2025-10-30

**Soundness:** 2
**Presentation:** 2
**Contribution:** 2
**Rating:** 2
**Confidence:** 4

**Summary:**

This paper proposes an efficient LLM architecture, namely recurrent transformer. The authors first consider the PAC-learning theory and claims that the theoretical-optimal model size should not be linearly scaling with the dataset size. More efficient architecture exists. Then they proposes the recurrent transformers, empirically validated over three toy tasks.

**Strengths:**

1. The good part of this research is the timely topic. Efficiency and scaling are central to current LLM research.
2. Recurrent transformer is lightweight, simple to implement, and tested on synthetic and small real datasets.

**Weaknesses:**

**On the theory side**:
1. The authors claims better scaling exists, however, the better scaling assumes the empirical fitting is true. The better scaling does not derive from the first principle, but accounting for the empirical scaling laws.
2. The theory doesn't direct connects to the proposed recurrent transformer architecture. Indeed, the architecture is seemingly directly combines the RNNs and Transformers.
3. The paper defines a discrete loss $\sum_{p_s \neq q_s} p_s$, treating probabilities as equal/not equal. This is atypical for PAC learning and breaks continuity assumptions needed for the generalization bounds it later invokes.

**On the empirical side**:
1. Scale mismatch. Experiments run on CIFAR-10, toy copy/selective-copy tasks, and nanoGPT Shakespeare. None validate large-scale efficiency claims; results are limited to very small models.
2. No comparisons. Extensive research on llm architectures proposed very strong baselines, such as mamba, deltaNet. However, this paper none of them, even the RNN.

**Novelty and Originality**:
I am not famililar with the line of research on LLM architectures, but the combination of RNN and Transformers is seemingly a easy-and-intuitive idea. I expect the authors to discuss the related works extensively.

**Questions:**

N/A

---

> ### Author Response · Authors · 2025-11-14
> **Thanks and Response to review**
>
> "weakness on the theory side"
>
>   (1) The authors claims better scaling exists, however, the better scaling assumes the empirical fitting is true. The better scaling does not derive from the first principle, but accounting for the empirical scaling laws.
>
> This is the scientific method for theoretical analysis of natural phenomena.  For example, the law of gravity and  the gravitational constant are derived from the analysis of experimental observations.  In our case, we extract a key property of natural language from large scale experiments published in the literature.
>
>   (2) The theory doesn't direct connects to the proposed recurrent transformer architecture. Indeed, the architecture is seemingly directly combines the RNNs and Transformers.
>
> The theory shows that efficient alternatives must exist, and the proposed architecture is one such.
>
>    (3)  The paper defines a discrete loss , treating probabilities as equal/not equal. This is atypical for PAC learning and breaks continuity assumptions needed for the generalization bounds it later invokes.
>
> The discrete loss function is the workhorse of PAC  learning and is the most widely used.
>
> On the empirical side:
> (1) Scale mismatch. Experiments run on CIFAR-10, toy copy/selective-copy tasks, and nanoGPT Shakespeare. None validate large-scale efficiency claims; results are limited to very small models.
>
> The paper is explicit that our experiments are resource constrained. No claims are made regarding larger models.
>
> (2) No comparisons. Extensive research on llm architectures proposed very strong baselines, such as mamba, deltaNet. However, this paper none of them, even the RNN.
>
> The paper does compare the recurrent transformer to RNNs etc.  For example, the proposed recurrent transformer of 0.5M parameters against the Mamba model of 100M parameters for Copy and Selective copy tasks.
>
> Novelty and Originality: I am not famililar with the line of research on LLM architectures, but the combination of RNN and Transformers is seemingly a easy-and-intuitive idea. I expect the authors to discuss the related works extensively.
>
> Agree that it would be challenging to review this paper absent familiarity with LLM architectures or PAC-learning theory.   Hope the updated version helps.
>
> Thanks

---

> ### Comment · Area_Chair_4GfW · 2025-11-23
>
> Dear reviewer,
>
> Thanks for your time and effort in reviewing ICLR2026 submissions. The authors have submitted their responses to your review. Please take the time to read and raise your further comments, and discuss with the authors.
>
> Best regards,
>
> AC

---

> ### Comment · Reviewer_6W5n · 2025-11-24
>
> Thank you for your response. Given the other reviewer's comment and the limited experimental scope, I am still lean to rejection.

---

> ### Author Response · Authors · 2025-11-28
> **Baffling**
>
> The reviewer claims a confidence level of 4: You are confident in your assessment, but not absolutely certain. It is unlikely, but not impossible, that you did not understand some parts of the submission or that you are unfamiliar with some pieces of related work.
>
> Yet states
> "Novelty and Originality: I am not famililar with the line of research on LLM architectures"
>
> Also states
> "The paper defines a discrete loss , treating probabilities as equal/not equal. This is atypical for PAC learning and breaks continuity assumptions needed for the generalization bounds it later invokes."
>
> The discrete loss is the workhorse of PAC learning and is most widely used.
>
> And:
> "The better scaling does not derive from the first principle, but accounting for the empirical scaling laws"
>
> The paper follows the scientific method for theoretical analysis of natural phenomena.

---

### Official Review · Reviewer_NvCo · 2025-11-02

**Soundness:** 2
**Presentation:** 3
**Contribution:** 2
**Rating:** 4
**Confidence:** 1

**Summary:**

The paper argues the “true” parameter–data scaling should grow sublinearly by tying empirical KL scaling to PAC-learning bounds, not linearly as in transformer-specific fits. It then proposes a recurrent transformer that reuses a single layer over a sliding window with a learnable memory knob to trade off forgetting vs. accumulation. The architecture claims linear-time processing in sequence length, better memory use, and plug-and-play batching while staying competitive in small-scale tests.

**Strengths:**

1. The sliding-window recurrence with a single reusable block is a clean, minimalist way to chase efficiency without throwing away attention.

2. The “learn to forget or accumulate” knob aligns with the intuition that language vs. long-range tasks want different memory behavior.

**Weaknesses:**

1. A single small window and one layer may miss cross-block interactions that deeper stacks capture implicitly.


2. The experiments run on modest hardware and narrow tasks, leaving open how this scales to modern pretraining or multi-billion-token corpora.

3. The memory knob’s behavior is shown qualitatively, but guidance on when it converges to “forget” vs. “accumulate” is thin.

**Questions:**

see weakness

---

> ### Author Response · Authors · 2025-11-14
> **Thanks and response to review**
>
> Regarding the three weaknesses raised in the review
>
> (1) As stated in the paper, we found no benefit to using additional layers.  We believe this might be because the recurrence inherently stacks successive blocks.
> (2)  While our experimental resources are limited, the theoretical bounds apply at any scale and we invite other researchers to build on the work.
> (3)  As stated in the paper,  the "memory knob" converged within a few batches in our experiments.
>
> All code available in the supplemental material for further experiments on ablation, convergence, scale etc.
>
> Thanks

---

> > ### Comment · Reviewer_NvCo · 2025-11-24
> >
> > Thanks for your reply. I share the same view as others.

---

> ### Comment · Area_Chair_4GfW · 2025-11-23
>
> Dear reviewer,
>
> Thanks for your time and effort in reviewing ICLR2026 submissions. The authors have submitted their responses to your review. Please take the time to read and raise your further comments, and discuss with the authors.
>
> Best regards,
>
> AC

---

### Author Response · Authors · 2025-11-14
**Thanks for the reviews**

Thanks for the reviews.

To our knowledge, this paper is the first to analyze the inherent complexity requirements of natural languages in the context of LLMs to derive a "natural AI scaling law."   We show that the popular transformer architecture and the related "AI scaling Law" are vastly over-provisioned in that the size of the model is linear in the size of the training data, while the natural requirement is roughly the square root of the size of the training data.   We then propose and experiment with an efficient alternative to transformers.  Our experiments are of limited scale owing to resource constraints.

We believe our theoretical results have substantial technical and business implications and suggest the reviewers weight its impact over the illustrative experiments that are suggestions for future work.  Attaching an update of the paper incorporating the review suggestions. Additional feedback and questions appreciated.

---

### Note · Authors · 2026-01-16

I have read and agree with the venue's withdrawal policy on behalf of myself and my co-authors.